Siderophore and indolic acid production by Paenibacillus triticisoli BJ-18 and their plant growth-promoting and antimicrobe abilities

Zhang Yunzhi 1
Ren Jinwei 2
Wang Wenzhao 2
Chen Baosong 2
Li Erwei liew@im.ac.cn 2
Chen Sanfeng chensf@cau.edu.cn 1
1 State Key Laboratory for Agrobiotechnology, College of Biological Sciences, China Agricultural University , Beijing , People’s Republic of China
2 State Key Laboratory of Mycology, Institute of Microbiology, Chinese Academy of Sciences , Beijing , People’s Republic of China
Silva Pedro
Electronic publication date: 2020 Jul 14
Publication date: 2020
Volume: 8
Electronic Location ID: e9403
Received 2019 Nov 8; Accepted 2020 Jun 1
Copyright: ©2020 Zhang et al.
Copyright year: 2020
Copyright holder: Zhang et al.
License: This is an open access article distributed under the terms of the Creative Commons Attribution License, which permits unrestricted use, distribution, reproduction and adaptation in any medium and for any purpose provided that it is properly attributed. For attribution, the original author(s), title, publication source (PeerJ) and either DOI or URL of the article must be cited.
License URL: https://creativecommons.org/licenses/by/4.0/

Keywords: Paenibacillus triticisoli BJ-18, Siderophore, Fusarinine, Indolic acid

Funding: National Key Research and Development Program of China 2017YFD0201705 Guangdong Innovative and Entrepreneurial Research Team Program 2013S033 This research was funded by the National Key Research and Development Program of China (grant number 2017YFD0201705) and by the Guangdong Innovative and Entrepreneurial Research Team Program (grant number2013S033). The funders had no role in study design, data collection and analysis, decision to publish, or preparation of the manuscript.

==============================
Paenibacillus triticisoli BJ-18, a N2-fixing bacterium, is able to promote plant growth, but the secondary metabolites that may play a role in promoting plant growth have never been characterized. In this study, untargeted metabolomics profiling of P. triticisoli BJ-18 indicated the existence of 101 known compounds, including N2-acetyl ornithine, which is the precursor of siderophores, plant growth regulators such as trehalose 6-phosphate, betaine and trigonelline, and other bioactive molecules such as oxymatrine, diosmetin, luotonin A, (-)-caryophyllene oxide and tetrahydrocurcumin. In addition, six compounds were also isolated from P. triticisoli BJ-18 using a combination of silica gel chromatography, sephadex LH-20, octadecyl silane (ODS), and high-performance liquid chromatography (HPLC). The compound structures were further analyzed by Nuclear Magnetic Resonance (NMR), Mass Spectrometry (MS), and Electronic Circular Dichroism (ECD). The six compounds included three classical siderophore fusarinines identified as deshydroxylferritriacetylfusigen, desferritriacetylfusigen, and triacetylfusigen, and three indolic acids identified as paenibacillic acid A, 3-indoleacetic acid (IAA), and 3-indolepropionic acid (IPA). Both deshydroxylferritriacetylfusigen and paenibacillic acid A have new structures. Fusarinines, which normally occur in fungi, were isolated from bacterium for the first time in this study. Both siderophores (compounds 1 and 2) showed antimicrobial activity against Escherichia coli, Staphylococcus aureus and Bacillus subtilis, but did not show obvious inhibitory activity against yeast Candida albicans, whereas triacetylfusigen (compound 3) showed no antibiosis activity against these test microorganisms. Paenibacillic acid A, IAA, and IPA were shown to promote the growth of plant shoots and roots, and paenibacillic acid A also showed antimicrobial activity against S. aureus. Our study demonstrates that siderophores and indolic acids may play an important role in plant growth promotion by P. triticisoli BJ-18.

Introduction

Plant growth-promoting bacteria (PGPB) have great usage as agricultural inoculants, such as biofertilization and biocontrol of pathogens (Backer et al., 2018). Commercialized PGPB strains mainly include the members of Agrobacterium, Azospirillum, Bacillus, Paenibacillus, Pseudomonas, Streptomyces et al. (Lucy, Reed & Glick, 2004; Banerjee et al., 2006). PGPB may promote plant growth directly usually by either facilitating resource acquisition (e.g., nitrogen fixation, production of indolic compounds and siderophores, phosphate solubilization, and 1—aminocyclopropane—1—carboxylate deaminase activity) or indirectly by decreasing the inhibitory effects of various pathogenic agents on plant growth and development (antibiotics and lytic enzymes); that is, by acting as biocontrol bacteria (Glick, 2012).

Nitrogen (N) fixation is catalyzed by molybdenum-dependent nitrogenase, which is a metalloenzyme composed of two protein components, referred to as MoFe protein and Fe protein. The atmospheric N2 is reduced to bioavailable NH4+ by nitrogen fixation, providing a large amount of natural N into cultivated agricultural systems (Galloway et al., 2008). In addition to symbiotic N2-fixing Rhizobia associated with legumes, the non-symbiotic N2-fixing bacteria are also important contributors to the N nutrition of non-legumes (Gupta, Roper & Roget, 2006). It is estimated that the microbial N accounts for roughly 30–50% of the total N in crop fields (Liu et al., 2017).

Iron is the fourth most abundant element on earth but its bioavailability is extremely limited due to its poor solubility (Braun & Braun, 2002; Miethke & Marahiel, 2007; Kazamia et al., 2018). Microbes have evolved strategies to obtain sufficient amounts of iron, such as chelation, reduction, and protonation (Guerinot, 1994). The use of a siderophore to transport iron by chelation is vitally important for bacteria. Siderophores are low molecular weight compounds (500–1,500 Da) possessing a high affinity for Fe3+ (Kf > 1030) (Haas & Defago, 2005) and they are synthesized by bacteria through non-ribosomal pathways (Hutchins et al., 1999; Khan et al., 2020). Many PGPB have been reported to produce siderophores, such as Bacillus subtilis, Paenibacillus polymyxa SK1, Mesorhizobium sp., Brevibacillus brevis GZDF3 (Franco-Sierra et al., 2020; Khan et al., 2020; Menéndez et al., 2020; Sheng et al., 2020). The chelate of siderophores and ferric iron can be directly absorbed by plants and are called mechanism III, which is thought to be used by plants to resist iron stress (Shenker et al., 1992; Yehuda et al., 1996; Chen, Dick & Streeter, 2000). When grown under iron-limiting conditions, mung bean plants by inoculation with the siderophore-producing Pseudomonas strain GRP3 showed reduced chlorotic symptoms and an enhanced chlorophyll level compared to uninoculated plants (Sharma et al., 2003). Tomato seedlings inoculation with these two Mesorhizobium strains that produce siderophores and IAA showed significantly higher plant growth traits than uninoculated seedlings (Menéndez et al., 2020). In addition, siderophores secreted by PGPB can suppress plant pathogens by competing for trace amounts of iron in the environment (Glick, 2012). It has been suggested that biocontrol PGPB produce siderophores that have a much greater affinity for iron than do fungal pathogens so that the fungal pathogens are unable to proliferate in the root rhizosphere of the host plant due to lack of iron (Schippers, Bakker & Bakker, 2003; O’Sullivan & O’Gara, 1992). For examples, siderophores have biocontrol roles against plant pathogens, such as Pirycularia oryzae (Yuquan et al., 1999), Stagonospora curtisii (Shuangya, Yongxiang & Xiangqun, 2003), Fusarium oxysporum (Duijff et al., 1993), Macrophomina phaseolina (Arora, Kang & Maheshwari, 2001) and Cryphonectria parasitica (Chen et al., 2006).

Indolic acids include several compounds, of which indole-3-acetic acid (indole acetic acid, IAA) is by far the most common as well as the most studied auxin (Xie et al., 2005). They stimulate cell division and promote cell elongation (Katzy et al., 1990; Weyers & Paterson, 2001). They are abundant in higher plants and rhizospheric microorganisms and play a vital role in plant–microbe interactions (Beneduzi et al., 2008; Costacurta & Vanderleyden, 1995; Lambrecht et al., 2000; Beck, Hansen & Lauritsen, 2003; Mao et al., 2014). Previous studies have demonstrated that Paenibacillus spp. can produce indolic compounds to promote plant growth (Kumari & Thakur, 2018; Castellano et al., 2018; Lebuhn, Heulin & Hartmann, 1997). Endophytic bacteria promote growth of the medicinal legume Clitoria ternatea by IAA production, P and K-solubilization (Aeron, Maheshwari & Meena, 2020).

Paenibacillus triticisoli BJ-18 (=DSM 25425T = CGMCC 1.12045T) is a N2-fixing bacterium isolated by our laboratory from wheat rhizosphere soil in Beijing (Wang et al., 2013). Recently, we have shown that inoculation with P. triticisoli BJ-18 significantly promotes the growth of tomato, maize and wheat (Xie et al., 2016; Shi et al., 2016; Li et al., 2019). The 15N-isotope-enrichment experiment indicated that plant seedlings inoculated with P. triticisoli BJ-18 derived 12.9–36.4% N from nitrogen fixation (Li et al., 2019). However, the secondary metabolites of P. triticisoli BJ-18 have never been isolated or characterized. In this study, six compounds, composed of three siderophores and three indolic acids, were isolated and characterized from P. triticisoli BJ-18. Notably, a new siderophore and a new indolic acid were here identified. Our results will provide insight into the mechanisms by which P. triticisoli BJ-18 promotes plant-growth, including nitrogen fixation and the secretion of siderophores and indolic acids.

Materials & Methods

Bacterial strain

The bacterial strain Paenibacillus triticisoli BJ-18 (=DSM 25425T = CGMCC 1.12045T) was used in our study.

Untargeted metabolomics by LC-MS

The equipment and raw data for untargeted metabolomics were provided by Beijing Novogene Technology Co., Ltd. P. triticisoli BJ-18 was cultured in Lysogeny Broth (LB) (10 g tryptone, 5 g yeast, 10 g NaCl, 15 g agar per 1 L H2O) at 30 °C for 3 days. A single colony was inoculated in 20 mL CH medium (30 g sucrose, 6.4 g tryptone, 7 g yeast, 0.6 g MgSO4 ⋅ 7H2O, 3.5 g NaCl, 0.1 g K2HPO4, 0.4 g KH2PO4 per 1 L H2O) and cultured at 30 °C 200 rpm for 48 h. The seed solution was inoculated into 150 mL of CH medium at an inoculation amount of 2% and cultured at 30 °C 200 rpm for 48 h. Cells were harvested by centrifugation at 6,000 rpm at 4 °C for 10 min, ground with liquid nitrogen, and resuspended with 500 µL of 80% methanol solution containing 0.1% formic acid. The homogenate was incubated on ice for 5 min and was centrifuged at 15,000 rpm at 4 °C for 10 min. 200 µL of supernatant was subsequently transferred to a fresh Eppendorf tube with a 0.22 µM filter and was centrifuged at 15,000 rpm at 4 °C for 10 min. LC-MS/MS analyses were performed using the Vanquish UHPLC system (Thermo Fisher) coupled with an Orbitrap Q Exactive series mass spectrometer (Thermo Fisher) in a Hyperil Gold column (100 × 2.1 mM, 1.9 µM) with a 16-min linear gradient at a flow rate of 0.2 mL/min. The eluents for the positive polarity mode were eluent A (0.1% formic acid in water) and eluent B (methanol). The eluents for the negative polarity mode were eluent A (5 mM ammonium acetate, pH 9.0) and eluent B (methanol). The solvent gradient was set as follows: 2% B, 1.5 min; 2–100% B, 12.0 min; 100% B, 14.0 min;100–2% B, 14.1 min; 2% B, 16 min. The Q exactive mass series spectrometer was operated in positive/negative polarity mode with spray voltage of 3.2 kV, sheath gas flow rate of 35 arb, aux gas flow rate of 10 arb, and a capillary temperature of 320 °C. Compound discoverer 3.0 (CD 3.0, Thermo Fisher) was used for processing the data files to match results from the mzCloud (https://www.mzcloud.org/) and ChemSpider (http://www.chemspider.com/) databases. Accurate qualitative and relative quantitative results were obtained through statistical analysis.

Reagents and instruments for separation and purification of compounds 1-6

HPLC data were acquired using a Waters 2695 instrument. HRESIMS and HPLC-ESI-MS data were obtained using an Accurate-Mass-Q-TOF LC/MS 6520 instrument (Agilent, USA) in positive ion mode. 1H and 13C NMR data were obtained by Bruker Avance-500 spectrometers (Rheinstetten, Germany) with solvent signals (DMSO-d 6, δH 2.500/ δC 39.520) as references, and HSQC and HMBC experiments were optimized for 145.0 and 8.0 Hz, respectively. The optical rotations measurement was conducted on an Anton Paar MCP200 polarimeter (Anton Paar, Austria). Optical rotations were conducted using a Perkin-Elmer 241 polarimeter, and UV data were detected by a Shimadzu Biospec-1601 spectrophotometer. The absorbance of 96-well clear plate was measured on a microplate reader (Molecular Devices, SpectraMax® Paradigm®). Analytically pure solvents including methanol, dichloromethane, and ethyl acetate were used for extraction and chromatographic separation. TLC was carried out on silica gel HSGF254 plates and spots were visualized by UV at 254 nm or sprayed with 10% H2SO4 and heated. Silica gel (150 −250 µM, Qingdao Haiyang Chemical Co., Ltd.), octadecylsilyl (ODS, 50 µM, YMC CO., LTD), and Sephadex LH-20 (Amersham Biosciences) were used for column chromatography (CC). HPLC separation was performed on an Agilent 1200 HPLC system with a Diode Array Detector (DAD) detector using an ODS column (C18, 250 × 9.4 mM, YMC Pak, 5 µM) at a flow rate of 2.0 mL/min.

Extraction and isolation of the secondary metabolites

P. triticisoli BJ-18 was grown in a 500 L fermenter (Biotech, Shanghai Baoxing Biological Equipment Engineering Co., Ltd.) filled with 200 L of CH medium at pH 5.0 with 40% to 60% dissolved oxygen and shaking at 100 rpm and at 30 °C for 48 h. For doing this, this bacterium was inoculated into 20 mL of CH medium and then scaled-up to 20 L. The seed culture was inoculated into 200 L of CH medium with an inoculation amount of 1% in a 500 L fermenter.

The fermented culture was passed through a 200 mesh macroporous adsorption resin following 3 periods of 30-minute ultrasonic cell-breaking and was eluted with analytically pure methanol. The organic solvent was collected and evaporated using rotary evaporators to obtain the crude extract (7 g). The aqueous phase was discarded after being extracted and separated three times between the ethyl acetate and the aqueous phases, and the organic phase was retained and evaporated using rotary evaporators. The final product was 4 g of EtOAc (ethyl acetate) extract.

The EtOAc extract was separated into 20 subfractions (Pt-1 to Pt-20) after being subjected to ODS column chromatography with a gradient of methanol-water (5–100%). The fraction Pt-10 was further partitioned by Sephadex LH-20 CC separated by 50% methanol in water to create 26 subfractions (Pt-10-1 to Pt-10-26). Compound 4 (4.2 mg, tR 31.2 min) was obtained from fraction Pt-10-24 (21.3 mg) by RP-HPLC using 55% methanol in acidic water. Fraction Pt-13 was divided into 16 fractions, Pt-13-1 to Pt-13-16, after being subjected to ODS column chromatography with a gradient of MeOH-H2O (40–80%). Compounds 1-3 (2 mg, 3.1 mg and 8 mg, tR 62.5 min, 58.4 min and 37.4 min, respectively) were obtained from fractions Pt-13-6 (42.5 mg) by RP-HPLC using 20% acetonitrile in acidic water. Fraction Pt-15 was further partitioned by Sephadex LH-20 CC eluted with 90% methanol in water to cut to 6 subfractions (Pt-15-1 to Pt-15-6). Compound 5 (214 mg, tR 42.5 min) was obtained from fraction Pt-15-4 (830 mg) by RP-HPLC using 72% methanol in acid water. Compound 6 (37.6 mg, tR 62.5 min) was isolated from fraction Pb-17 (928 mg) by RP-HPLC (C8) using 80% methanol in acid water.

Deshydroxylferritriacetylfusigen (compound 1): yellow powder; [α]25D = −14.0 (c 0.10, MeOH); UV (MeOH) λmax (log ε) 214 (2.44) nm; NMR data (500 MHz, DMSO-d6) is shown in Table 1; positive HRESIMS m/z 837.4242 [M + H]+ (calculated for C39H61N6O14, 837.4246 Δ = 0.0004).

Table 1 NMR data of compound 1 in DMSO-d6.

Pos.	δCa	δHb, mult (J in Hz)	
1	166.0		
2	117.5	6.31, s	
3	150.0		
4	31.9	2.80, br. s	
5	63.1	4.17, m	
		4.12, m	
6	172.1		
7	51.8	4.19, m	
8	28.0	1.65, m	
9	23.0	1.53, m	
10	46.3	3.51 br. s	
11	25.3	1.87, s	
12	169.5		
13	22.3	1.84, s	
	N-OH	9.70, br. s	
	N-H	8.20, d (7.5)	
Notes.

a Recorded at 125 MHz.

b Recorded at 500 MHz.

Desferritriacetylfusigen (compound 2): yellow powder; [ α]25D = −8.0 (c 0.05, MeOH); UV (MeOH) λmax (log ε) = 214 (2.43) nm; positive HRESIMS m/z 853.4203 [M + H]+, (calculated for C39H60N6O15, 853.4195 Δ = 0.0008).

Triacetylfusigen (compound 3): brown powder; [α]25D = −216.0 (c 0.025, MeOH); UV (MeOH) λmax (log ε) 211 (2.35) nm; positive HRESIMS m/z 906.3316 [M + H]+ (calculated for C39H58N6O15Fe, 906.3309 Δ = 0.0007).

Paenibacillic acid A (compound 4): white powder; [α]25D= −39.99 (c 0.10, MeOH); UV (MeOH) λmax (log ε) 211 (2.33) nm; NMR data (500 MHz, DMSO-d 6) is shown in Table 2; positive HRESIMS m/z 303.1704 [M + H]+ (calculated for C17H23N2O3 303.1708 Δ = 0.0004).

Table 2 NMR data of compound 4 in DMSO-d6.

Pos.	δCa	δHb, mult (J in Hz)	
1	39.5	4.35, d (15.7)	
		4.21, d (15.7)	
3	56.1	3.63, dd (10.3, 5.0)	
4	22.9	3.13, dd (15.3, 5.0)	
		2.84, dd (15.3, 10.4)	
4a	108.2		
4b	126.2		
5	118.1	7.47, d (7.5)	
6	119.7	7.07, t (7.5)	
7	121.9	7.17, t (7.5)	
8	109.8	7.55, d (7.5)	
8a	137.2		
9a	129.4		
1′	170.0		
2′	72.3	5.46, s	
3′	67.1	3.30, m	
4′	31.0	1.40, m	
5′	18.8	1.22, m	
6′	13.6	0.78, t (7.4)	
Notes.

a Recorded at 125 MHz.

b Recorded at 500 MHz.

3-Indoleacetic acid (IAA) (compound 5): white amorphous powder; UV (MeOH) λmax (log ε) 234 (4.22), 272 (3.77), 280 (3.80), 290 (3.72) nm; positive HRESIMS m/z 176.0709 [M + H]+ (calculated for C10H10NO2 176.0701 Δ = 0.0008).

3-Indolepropionic acid (IPA) (compound 6): white amorphous powder; UV (MeOH) λmax (log ε) 235(4.25), 270 (3.77), 280 (3.80), 290 (3.72) nm; positive HRESIMS m/z 190.0871 [M + H]+ (calculated for C11H12NO2, 190.0868 Δ = 0.0003).

Conversion of compound 3 into desferritriacetylfusigen

The removal of the ferric ion was conducted as reported by Kodani et al. (2015) with additional changes. Compound 3 (4 mg) was dissolved in 1.5 mL of water and then 1.5 mL of 1M 8-quinolinol was added. The solution was stirred for 25 min and mixed with three mL of CH2Cl2, then left to stand until bilayer separation occurred. The water layer was evaporated using a freeze-dryer vacuum. This was repeated 3 times to eliminate ferri-8-quinolinol and obtain the total dry material. The sample was then dissolved in 100 µL of methanol and HPLC purification was performed with the C18 column (7.8 × 300 mM, Waters, 7 µM; detector: UV), eluted with 20% acetonitrile in water at a flow rate of one mL/min, and monitored at OD210 to create 2.0 mg of desferri-compound 3. Positive HRESIMS and NMR of desferri-compound 3 was the same as compound 2.

ECD computations

Systematic conformation analysis of 4a were conducted with CONFLEX (version 7 Rev. A; CONFLEX Corporation) using the MMFF94 molecular mechanics force field. Optimization with DFT calculation at the B3LYP/6-31G(d) level in MeOH in methanol with the PCM model using the Gaussian 09 program (Revision C.01, Gaussian Inc.) afforded the MMFF minima. The 40 lowest electronic transitions were calculated using time-dependent density-functional theory (TDDFT) methodology at the B3LYP/6-31G(d) level. The overall ECD spectra were then produced based on Boltzmann weighting of each conformer as described in literature(Liu et al., 2018a; Liu et al., 2018b).

Determination of plant growth promoting capacity of indolic compounds

The seeds of Arabidopsis thaliana var. Columbia were sterilized using 75% ethanol for 45 s, 5% NaClO for 15 min, and were washed in sterile water 3 times. The seeds were then placed horizontally on solid 1/2 Murashige and Skoog (MS) (Murashige & Skoog, 1962) medium supplemented with 1.5% sucrose, 0.35% (w/v) agar, and different concentrations and combinations of 1 mg/L paenibacillic acid A, IAA, and IPA for direct regeneration. At least ten Arabidopsis strains were cultured in each group of treatments. The cultures were incubated at 25 ± 2  °C under cool fluorescent light (2000 lux 16 h/day photoperiod) for 14 days. The ability of the three indolic acids to promote plant growth was evaluated by measuring the dry weight of the stems and roots of Arabidopsis thaliana.

Detection of indolic compounds

The Salkowski Reagent (PC technique) was used to determine concentrations of indolic compounds (Glickmann & Dessaux, 1995). IAA standard solutions with concentrations of 0, 5.0, 10.0, 15.0, 20.0, and 25.0 µg /mL were used to react with an equal volume of PC reagent in the dark for half an hour and the OD530 value was measured to draw the Salkowski calibration curve. The R2 value should be greater than 0.99.

P. triticisoli BJ-18 was grown overnight in 50 mL of LB medium at 200 rpm 30 °C. The cells were collected by centrifugation, washed three times with deionized water, and resuspended in one mL of deionized water. 100 µL of the bacterial suspension was added to 20 mL nitrogen-deficient medium (prepared with deionized water and supplemented with 100 mM NH4Cl as a nitrogen source) with 36 mg/L iron citrate or no iron citrate. After three days of incubation at 200 rpm and 30  °C, all samples were treated with the same method to measure the corresponding OD530 value. The content of indolic compounds of each sample was calculated using the Salkowski calibration curve. The nitrogen-deficient medium with deionized water was used as a blank control. Each sample was treated three times.

Antimicrobial assay

The antimicrobial assay was performed according to the recommended guidelines of the National Center for Clinical Laboratory Standards (NCCLS) (Li et al., 2008). The bacterial cells of Escherichia coli (ATCC1.0090), Staphylococcus aureus (ATCC6538), and Bacillus subtilis (ATCC6663) were grown in Mueller-Hinton Broth (MHB) medium (5.0 g beef powder, 1.5 g starch, 17.5 g acid hydrolyzed casein, and 1 L H2O) at 37 °C for 24 h. The fungus (yeast), Candida albicans (CGMCC2.2086), was grown in Saurer’s Dextrose Broth (SDB) medium (5 g casein trypsin digest, 5 g gastric enzyme digest, 20 g dextrose and 1 L ddH2O) at 28 °C for 48 h.

The cells of bacteria and yeast cultivated in MHB and SDB media as described above were seeded onto each well of a 96-well clear plate, then the gradient concentrations of test compounds 1-6 were added to each well and mixed with the targeted microbes to reach a final volume of 100 µL. The inoculation concentrations of bacteria and yeast were 0.5–2.5 × 105 cfu / mL and 0.5–2. 5 × 103cfu / mL, respectively. The suspension was cultured at 37 °C for 1 day and 28 °C for 2 days, respectively. Gentamicin, ampicillin, sulphate streptomycin, and amphotericin B were used as positive controls. A microplate reader was employed to perform at the OD595. The IC50 of compounds 1-6 were plotted, calculated, and obtained. Each antimicrobial assay was tested in triplicate.

Siderophores detection by blue agar CAS assay

Siderophores were detected using a blue agar CAS medium as described by Schwyn & Neilands (1987). The solid CAS medium was composed of one mL 20% sucrose solution, three mL sterilized 10% casamino acid, 100 µL one mmol/L CaCl2, five mL CAS dye solution (a mixture of 0.012 g of chrome azurol S in 10 mL of ddH2O, two mL one mmol/L FeCl3, and 0.015 g hexadecyltrimethylammonium bromide in eight mL of deionized H2O), PIPES buffer (pH 6.8–7.0), and 2 g agar powder per 100 mL H2O.

P. triticisoli BJ-18 was inoculated in 20 mL of liquid LB medium with shaking at 200 rpm at 30 °C for 48 h. one mL of the bacterial solution was centrifuged, washed with sterile deionized water, and then suspended with 200 µL of sterile deionized H2O. 10 µL of the suspension was inoculated on a CAS plate. The color changed from blue to light orange, indicating the presence of iron carriers.

Results

Untargeted metabolomics profiling of P. triticisoli BJ-18

The untargeted metabolomics profiling of P. triticisoli BJ-18 were analyzed by using LC-MS and the data were comparatively analyzed with the databases of mzCloud, ChemSpider and mzVault. A total of 101 compounds were measured, the majority of which were common compounds involved in the basic metabolism (Table S1), including carbohydrates, amino acids, peptides, alcohols, aldehydes, ketones, fatty acids, lipids, nucleic acids, vitamins, alkaloids, cyclics and their respective derivatives (Fig. 1A). Of the 101 compounds, 46 have the molecular weights of 100–200 Da, 34 have the molecular weights of 200–300 Da, 18 have the molecular weights of more than 300 Da and 3 have the molecular weights of less than 100 Da (Fig. 1B).

Figure 1 Untargeted metabolomics profiling of Ptriticisoli BJ-18.

(A) Number of compounds of different types; (B) Number of compounds in different molecular weight.

Notably, N2-acetyl ornithine, a precursor to fusarinines, is included among these compounds. Also, plant growth regulators such as trehalose 6-phosphate (T6P, resistant to drought and salt stress. Wingler et al., 2012; Prasad et al., 2014; Kretzschmar et al., 2015), betaine (a non-toxic osmotic regulator. (Cho et al., 2003; Tramontano & Jouve, 1997) and trigonelline (resistant to salt stress. Minorsky, 2002; Tramontano & Jouve, 1997), and other active molecules such as oxymatrine (a drug used to treat hepatitis B and tumors. Lu et al., 2003; Song et al., 2006), diosmetin (in food or medicine with anti-oxidant properties, anti-infective, and anti-shock functions Pallab et al., 2019), luotonin A (anti-tumor drug. González-Ruiz et al., 2010), α-humulene (hippone, sesquiterpene, with anti-inflammatory effect. Rogerio et al., 2009), (-)-caryophyllene oxide (anti-tumor and antifungal drug. Yang et al., 1999; Park et al., 2011), tetrahydrocurcumin (hepatotoxicity prevention drug and natural whitening ingredients. Pari & Murugan, 2004; Trivedi et al., 2017) were also included among these compounds.

Structural determination of secondary metabolites from P. triticisoli BJ-18

The cells of P. triticisoli BJ-18 were fermented and concentrated, and 4 g of EtOAc extracts was obtained. The extracts were separated using a combination of silica gel chromatography, sephadex LH-20, ODS column chromatography, and HPLC. Six compounds (compounds 1-6) were ultimately obtained and were further analyzed using NMR and MS to establish their structures. Further the structure and characters of the six compounds (compounds 1-6) were compared with the corresponding compounds in the literature. Among the six compounds, compounds 1-3 were identified as fusarinines that were classical siderophores, while compounds 4-6 were identified as indolic acids (Fig. 2). Both compound 1 and 4 have new structures.

Figure 2 Compounds isolated from P. triticisoli BJ-18.

Six compounds were isolated from P. triticisoli BJ-18, including three classical siderophore fusarinines identified as (A) deshydroxylferritriacetylfusigen, (B) desferritriacetylfusigen and (C) triacetylfusigen, and three indolic acids identified as (D) paenibacillic acid A, (E) 3-indoleacetic acid(IAA), and (F) 3-indolepropionic acid (IPA).

Compound 2 was a cyclic tripolymer which has three same monomers (m/z 284.1366) identified as desferritriacetylfusigen (Anke, 1977). It was characterized by 1H NMR (500 MHz, DMSO-d 6), δH 9.70 (s), 8.21 (d, J = 7.5 Hz), 6.32 (s), 4.30–4.03 (m, 3H), 3.33 (s), 2.81 (m), 1.88 (s), 1.85 (s), 1.75–1.39 (m, 4H) (Fig. S1) and by 13C NMR (125 MHz, DMSO-d 6) δC 22.7, 23.5, 25.8, 28.5, 32.4, 46.8, 52.3, 63.5, 117.9, 150.1, 167.4, 170.0, 172.6 (Fig. S2). Compound 5 was identified as 3-indoleacetic acid (IAA) (Gathungu et al., 2014). As shown in Fig. S2 , it was characterized by 1H NMR (500 MHz, DMSO-d6), δH 12.14 (s), 10.90 (s), 7.49 (d, J = 7.9 Hz), 7.34 (d, J = 8.1 Hz), 7.22 (d, J = 2.3 Hz), 7.07 (t, J = 7.6 Hz), 6.98 (t, J = 7.4 Hz), 3.63 (s, 2H). Compound 6 was identified as 3-indolepropionic acid (IPA) (Rustamova et al., 2019). As shown in Fig. S4, it was characterized by 1H NMR (500 MHz, acetone-d 6), δH 9.97 (s), 7.59 (d, J = 7.9 Hz), 7.37 (d, J = 8.1 Hz), 7.16 (s), 7.09 (t, J = 7.5 Hz), 7.02 (t, J = 7.5 Hz), 3.06 (t, J = 7.7 Hz), 2.70 (t, J = 7.7 Hz).

Compound 1 was a new member of fusarinines (siderophores) and compound 4 was a new member of the indolic acids. Their structures were further determined by extensive spectroscopic experiments. The structure of compound 3 was determined by NMR after iron was removed and it was identified as triacetylfusigen. Overall, the six compounds include three fusarinines (siderophores) and three indolic acids. (Fig. 2).

Compounds 1, 3, and 4, are described in greater detail as follows:

Deshydroxylferritriacetylfusigen (compound 1) was a yellow powder with a molecular formula of C39H60N6O14 (thirteen degrees of unsaturation) by HRESIMS m/z 837.4242 [M + H]+ (calculated for C39H61N6O14 , 837.4246). Its 1H, 13C and HSQC NMR spectroscopic data (Figs. S5– S7) suggested the existence of two methyl groups (δC∕H22.3/1.84 and 25.3/1.87), five methylenes, including one O-methylene (δC∕H63.1/4.17 and 4.12), one double bond (δC∕H117.5/6.31 and δC 149.6), three carboxylic carbons (δC 166.0, 169.5 and 172.1), one N-methine (δC∕H 51.8/4.19), one amide N-H proton (δH 8.23), and one amide N-OH proton (δH9.92). Thirteen carbon signals in the 13C NMR and thirty-nine in the molecular formula indicated that compound 1 was a cyclic tripolymer similar to compound 2. Comparison of MS-MS spectrum of compound 1 (Fig. S8) and compound 2 (Fig. S9) revealed that unlike compound 2, two of the three monomers of compound 1 have the same molecular weight as the monomers of compound 2 (m/z 284.1366), and the other was different (m/z 268.1566). The molecular weight of this unique monomer is 16 less than that of others, which is the molecular weight of an oxygen atom. Although the NMR of compound 1 was similar to compound 2, the integration of N-OH protons in 1H NMR of compound 1 was less than compound 2. The molecular formula also suggested that there were only two hydroxyls in compound 1 rather than three. The planar structure was confirmed by combining 1H-1H COSY and HMBC (Figs. S10 & S11, Fig. 3). The ROESY spectrum (Fig. S12) showed that CH3-11 had the NOE correlation with the olefinic proton (δH6.31) indicating that the double bond was in a Z- configuration. Compound 2 was deduced to come from ornithine (Schrettl et al., 2007), and the absolute configuration of compound 1 was determined to be 5S. Our data suggested that compound 1 has a new structure and we proposed the name deshydroxylferritriacetylfusigen for the new number of fusarinines (Table 1).

Figure 3 Key 1H-1H COSY and HMBC correlations of compounds 1 and 4.

1H-1H COSY spectra (DMSO-d6, 8 MHz); HMBC spectra (DMSO-d6, 8 MHz).

Compound 3 was a brown powder with a molecular formula of C39H57N6O15Fe by HRESIMS ([M + H]+ at m/z 906.3316, calculated for C39H58N6O15Fe, 906.3309). Compound 3 had no signal of NMR (Fig. S13) due to its metal-shielding characteristics. A strong complexing agent (Kodani et al., 2015) desferri-compound 3 was a yellow powder with a molecular formula of C39H60N6O15 (thirteen degrees of unsaturation) by HRESIMS ([M + H]+ at m/z 853.4203, calculated for C39H60N6O15, 853.4195) obtained following the precipitation and separation of the metal ions by 8-quinolinol. Analysis of its HRESIMS and 1H NMR (Fig. S14) data revealed that desferri-compound 3 had the same structure as compound 2. Compound 3 was determined to be a ferri-complex compound, identified as triacetylfusigen.

Compound 4 was isolated as white amorphous powder with the molecular formula of C17H22N2O3 (eight degrees of unsaturation) determined by HRESIMS m/z 303.1704 [M + H]+ (calculated for C17H23N2O3 303.1708). The 1H and 13C NMR combining with HSQC (Figs. S15– S17) revealed one methyl group [δC∕H13.6/0.78 (t, J = 7.4Hz)], six methylenes, including two O-methylene [δC∕H 67.1/3.30 (m) and 72.3/5.46 (s)], one ethane [δC 56.1/3.63 (dd, J = 10.3, 5.0)], eight aromatic/olefinic carbons (δC 108.2, 109.8, 118.1, 119.7,121.9, 126.2, 129.4 and 137.2) and one carboxylic carbons (δC170.0). These data suggested that compound 4 was similar to lycoperodine-1 (Yahara et al., 2004). The 1H–1H COSY data (Fig. S18) revealed four isolated proton spin-systems of C-1-N-C-3-C-4, C-5–C-6, C-7–C-8, and C-3′–C-4′–C-5′–C-6′, respectively. The HMBC spectrum (Fig. S19) showed correlations from H-2′ to C-3′, C-8a and C-9a, from H-1 to C-3 and C-9a, from H-3 and H-4 to C-1, from H-4 to C-9a and C-4a, from H-5 to C-5a and C-4a, and from H-7 and H-8 to C-8a. The planar structure of compound 4 was established by linking these fragments (Fig. 3 & Table 2). The ECD calculation method (Ma et al., 2014) was used to determine the absolute configurations. The structure of compound 4 was simplified into two stereoisomers 4a (3S) and 4b (3R). The calculated ECD curve of compound 4a correlated with the experimental CD spectrum of 4 (Fig. 4). The absolute configuration of compound 4 was established as 3S using the time-dependent density functional theory (TDDFT) at the B3LYP/6-311G (d,p) level (Fig. S20). Compound 4 was the derivative of IPA (here named paenibacillic acid A).

Figure 4 Experimental ECD spectra of 4 and its calculated ECD spectra of related simplified possible stereoisomers 4a and 4b.

Experimental ECD spectra of 4 was recorded in methanol. Systematic conformation analysis of 4a was conducted with CONFLEX using the MMFF94 molecular mechanics force field. Optimization with DFT calculation at the B3LYP/6-31G(d) level in MeOH by the Gaussian09 program afforded the MMFF minima. At the B3LYP/6-31G(d) level, the exciting states were calculated using time-dependent density-functional theory (TDDFT) methodology for 4a. The overall ECD spectra were then produced based on Boltzmann weighting of each conformer.

Figure 5 Plant growth-promoting capacity of indolic compounds on Arabidopsis thaliana.

Seeds of Arabidopsis thaliana var. Columbia were used as plant material, and were incubated at 25 ± 2 °C under cool fluorescent light (2000 lux 16 h/day photoperiod) for 14 days.

Figure 6 Comparison of plant growth-promoting ability of indolic compounds.

At least ten Arabidopsis strains were cultured in each group of treatments (A, paenibacillic acid A (nM); B, 3-indoleacetic acid (nM); C,3-indolepropionic acid (nM)). The cultures were incubated at 25 ± 2  °C under cool fluorescent light (2000 lux 16 h/day photoperiod) for 14 days.

Effects of indolic acids compounds on plant growth promotion

The effects of indolic acids compounds on growth of Arabidopsis thaliana var. Columbia were investigated. Paenibacillic acid A, IAA, and IPA were shown to promote the growth of plant shoots and roots (Fig. 5) as well as the dry weight of the plants (Fig. 6). The growth-promoting ability increased as the concentration of paenibacillic acid A, IAA, and IPA increased until their highest growth-promoting ability was attained, and then their plant growth-promoting ability decreased with increasing concentration. The significant efficiencies of plant-growth promotion were obtained when the concentrations of paenibacillic acid A, IAA and IPA were at 100 nM, 250 nM and 50 nM, respectively, suggesting that IPA has the strongest ability of plant growth promotion among the three indolic acids compounds.

We further determined the content of indolic compounds produced by P. triticisoli BJ-18 when it was measured in the nitrogen-deficient medium with and without iron. The results showed that P. triticisoli BJ-18 produced 37.03 ± 1.21 µg/mL of the indolic compounds in a medium with an iron ion concentration of 4.409 × 10−4 mM and 13.457 ± 0.78 µg/mL of the indolic compounds in a medium without iron, suggesting that the content of indolic compounds in P. triticisoli BJ-18 is positively related to the iron in the environment.

Antimicrobial activity

The antimicrobial properties of siderophores and indolic acids were assayed against the indicator strains (bacteria E. coli, S. aureus and B. subtilis, and yeast C. albicans) (Table 3). Both compounds 1 and 2 that are siderophores showed antimicrobial activity against E. coli, S. aureus and B. subtilis, but did not show obvious inhibitory activity against yeast C. albicans.

Gentamicin, ampicillin, streptomycin sulfate and amphotericin B were used as positive controls against E. coli, S. aureus, B. subtilis and C. albicans, respectively. The compounds were tested at concentrations of 5 µM, 10 µM and 20 µM. The IC50 was calculated using the Spearman-Karber’s method.

Table 3 Antimicrobial activity of compounds 1-6.

Gentamicin, ampicillin, streptomycin sulfate and amphotericin B were used as positive controls against E. coli, S. aureus, B. subtilis and C. albicans, respectively. The compounds were tested at concentrations of 5 µm , 10 µm and 20 µm. The IC50 was calculated using the Spearman-Karbers method. The horizontal line “–” indicated that the compound had no antibacterial activity against the indicator strain.

IC50 (µM)	E. coli	S. aureus	B. subtilis	C. albicans	
compound 1	6.8	5.2	8.8	–	
compound 2	4.7	4.3	6.4	–	
compound 3	–	–	–	–	
compound 4	–	7.4	–	–	
compound 5	–	–	–	–	
compound 6	–	–	–	–	
Pos.	1.9	2.1	2.3	2.2	

Siderophore detection by blue agar CAS assay

Siderophore production by P. triticisoli BJ-18 was determined by blue agar CAS assay as described in materials and methods. A yellow ring appeared around each colony after seven days of P. triticisoli BJ-18 growth on a blue agar plate, indicating that P. triticisoli BJ-18 had the ability to secrete siderophores and transfer iron from the environment to the bacterial cells (Fig. S21). The data are consistent with the above results that compounds 1-3 are siderophores.

Discussion

P. triticisoli BJ-18, a N2-fixing bacterium, significantly promoted plant growth, but the secondary metabolites produced by this bacterium have never been characterized. In this study, 101 known compounds of P. triticisoli BJ-18 were measured by untargeted metabolomics profiling. These compounds include N2-acetyl ornithine, which is the precursor of fusarinines (siderophores). There are 7 types of siderophores: fusarinines, rhodotorulic acids, ferrichromes, ferrioxamines, aerobactins, enterobactins, and mycobactins (Hossain et al., 1980). Fusarinines are synthesized by aminoacyl bonds, making them different from other types of siderophores that are polymerized using peptide linkages. Fusigen, a cyclic trimester of fusarinine, was identified by Diekmann & Zähner (1967) and was considered to be the iron ionophore for Fusar ium roseum (Sayer & Emery, 1968). Studies have shown triacetylfusigen, which has three acetyl group in place of the H atoms of the amino, was isolated from Aspergillus fumigates (Diekmann & Krezdorn, 1975) and Aspergillus nidulans (Charlang et al., 1981), while desferritriacetylfusigen was found in Aspergillus deflectus (Anke, 1977) and Emericella sp. (Cruz et al., 2012). These siderophores were isolated from fungi but bacterial fusarinines have never been identified before our study. Here, three fusarinines were isolated from P. triticisoli BJ-18 and identified as deshydroxylferritriacetylfusigen, desferritriacetylfusigen, and triacetylfusigen, of which deshydroxylferritriacetylfusigen was a new structure of fusarinine. Here is the first study to report the bacterial fusarinine. Ornithine is the only amino acid of fusarinine (Charlang et al., 1982). The Kyoto Encyclopedia of Genes and Genomes (KEGG, https://www.kegg.jp/) shows that fusarinine can be synthesized by some specific ornithylesterases including fusarinine-C ornithineesterase, ornithine esterase and 5-N-acyl-L-ornithine-ester hydrolase. Studies on fusarinine in fungi have shown that the specific ornithylesterases promote cellular iron-exchange by hydrolysis of the ester bonds of the ferric ionophores (Emery, 1976).

Indolic compounds are natural auxins in plants and rhizosphere microorganisms and can promote the formation of shoot tips, buds, and roots. The auxins commonly used in agriculture are IAA, indolebutyric acid, 2,4-dichlorophenoxyacetic acid and naphthylacetic acid. Three indolic acids identified as paenibacillic acid A, IAA, and IPA were isolated from P. triticisoli BJ-18, of which paenibacillic acid A was a new structure. The results of the plant growth promoting capacity assay showed that the three compounds can promote the growth of the shoots and roots of A. thaliana and can increase the dry weight of the plant. The effect of paenibacillic acid A on A. thaliana is similar to that of IAA. IPA was the best promoter of plant growth among the three indolic compounds. Indolic compounds such as indole, skatole, and indirubin were concurrently identified by untargeted metabolomics profiling. Our results showed that P. triticisoli BJ-18 can promote plant growth by synthesizing plant growth hormones, including indolic acids.

Indolic compounds were usually synthesized in two pathways, either by iron (III) complexed by the ligand of indolic compounds or by the reduction of the soluble iron (II) complex. These parallel reactions cannot proceed without iron (Gazaryan et al., 1996;Kovács et al., 2008; Xie et al., 2016). Indolic acid assays showed that the content of indolic compounds of P. triticisoli BJ-18 was related to the presence or absence of iron ions. The content of indolic compounds of P. triticisoli BJ-18 can reach 37.03 ± 1.21 µg/mL in the medium with an iron ion concentration of 4.409 × 10−4 mM, while the content is 13.457 ± 0.78 µg/mL in the iron-free environment, indicating the importance of iron for the synthesis of indolic compounds.

Phytopathogens must sequester iron to develop and sustain infections (Ratledge & Dover, 2000). Competing for iron is a mechanism taken by PGPR to inhibit the growth of phytopathogens in the soil (Chet et al., 1990). The six compounds were evaluated for their activities against a panel of microbes. Results showed that desferritriacetylfusigen and deshydroxylferritriacetylfusigen were antimicrobial against E. coli, S. aureus, and B. subtilis. Triacetylfusigen showed no antibiosis activity against any targeted microorganism due to its complexion with iron.

Plant growth regulators such as betaine and trigonelline, and other active molecules such as luotonin A, aphidicolin, oxymatrine, diosmetin, pilocarpine and tetrahydrocurcumin were also identified by untargeted metabolomics profiling. Besides, deshydroxylferritriacetylfusigen had weak cyotoxic activity against AsPC-1 with IC50 value of 81.2 ± 3.9 µM (File S1), indicating that siderophores may have potential as a new therapy for human cancers (Kalinowski & Richardson, 2005; Ji, Juárez-Hernández & Miller, 2012). The chemical composition and application of P. triticisoli BJ-18 is far more complicated and promising.

P. triticisoli BJ-18 has been shown to promote plant growth using several mechanisms (Fig. 7). P. triticisoli BJ-18 provides a nitrogen source for plant growth by N2-fixation, produces indolic acids to promote plant growth, generates siderophores to capture iron atoms to synthesize nitrogenase and indolic acids, and synthesizes plant growth regulators such as T6P, betaine and trigonelline, and secretes fusarinines and paenibacillic acid A to resist phytopathogens.

Figure 7 P. triticisoli BJ-18 promotes plant growth by several mechanisms.

The orange squares represent plant growth regulators such as T6P, betaine and trigonelline. The purple ring represents fusarinines and the red dot represents iron atoms. The blue triangle indicates indole acids. Red radial circles represent plant pathogens. Six-component structure representing nitrogenase.

Our results provide chemical evidence for the use of P. triticisoli BJ-18 as a PGPR biofertilizer in agriculture. The discovery of the many compounds identified in our study shows the agricultural and medical value of P. triticisoli BJ-18. The metabolic regulation of fusarinines and other bioactive compounds requires further study.

Conclusions

In this study, six compounds were isolated and characterized from P. triticisoli BJ-18, a N2-fixer. The six compounds included three classical siderophore fusarinines identified as deshydroxylferritriacetylfusigen, desferritriacetylfusigen, and triacetylfusigen, and three indolic acids identified as paenibacillic acid A, 3-indoleacetic acid (IAA), and 3-indolepropionic acid (IPA). Both deshydroxylferritriacetylfusigen and paenibacillic acid A have new structures. Fusarinines, which normally occur in fungi, were isolated from bacteria for the first time in this study. Both siderophores (compounds 1 and 2) showed antimicrobial activity against E. coli, S. aureus and B. subtilis, but did not show obvious inhibitory activity against yeast Candida albicans. Whereas triacetylfusigen (compound 3) showed no antibiosis activity against these test microorganisms. Paenibacillic acid A, IAA, and IPA were shown to promote the growth of plant shoots and roots, and paenibacillic acid A also showed antimicrobial activity against S. aureus. Our study demonstrated that siderophores and indolic acids may play an important role in plant growth promotion by P. triticisoli BJ-18.

Supplemental Information

Supplemental Information 1 The cytotoxicity bioassay of deshydroxylferritriacetylfusigen against AsPC-1

Click here for additional data file.

Supplemental Information 2 1H NMR spectrum of compound 2

DMSO-d _6, 500 MHz.

Click here for additional data file.

Supplemental Information 3 APT-13C NMR spectrum of compound 2

DMSO-d 6, 125 MHz.

Click here for additional data file.

Supplemental Information 4 1H NMR spectrum of compound 5

DMSO-d6, 500 MHz.

Click here for additional data file.

Supplemental Information 5 1H NMR spectrum of compound 6

DMSO-d6, 500 MHz.

Click here for additional data file.

Supplemental Information 6 1H NMR spectrum of compound 1

DMSO-d6, 500 MHz.

Click here for additional data file.

Supplemental Information 7 APT-13C NMR spectrum of compound 1

DMSO-d6, 125 MHz.

Click here for additional data file.

Supplemental Information 8 HSQC spectrum of compound 1

DMSO-d6, 145 MHz.

Click here for additional data file.

Supplemental Information 9 MS-MS spectrum of compound 1

MS-MS spectrum was detected under 130.0 V. Compound 1 has three subunits, two of which have the same molecular weight (m/z 284.1366) and the other was different (m/z 268.1566).

Click here for additional data file.

Supplemental Information 10 MS-MS spectrum of compound 2

MS-MS spectrum of compound 2 was detected under 130.0 V. Compound 2 has three subunits of the same molecular mass (m/z 284.1366).

Click here for additional data file.

Supplemental Information 11 1H-1H COSY spectrum of compound 1

DMSO-d6, 8 MHz.

Click here for additional data file.

Supplemental Information 12 HMBC spectrum of compound 1

DMSO-d6, 8 MHz.

Click here for additional data file.

Supplemental Information 13 ROESY spectrum of compound 1

DMSO-d6.

Click here for additional data file.

Supplemental Information 14 1H NMR spectrum of compound 3

CDCl3, 500 MHz (no signal except solvent).

Click here for additional data file.

Supplemental Information 15 1H NMR spectrum of desferri-compound 3

DMSO-d6, 500 MHz.

Click here for additional data file.

Supplemental Information 16 1H NMR spectrum of compound 4

DMSO-d6, 500 MHz.

Click here for additional data file.

Supplemental Information 17 APT-13C NMR spectrum of compound 4

DMSO-d6, 125 MHz.

Click here for additional data file.

Supplemental Information 18 HSQC spectrum of compound 4

DMSO-d6, 145 MHz.

Click here for additional data file.

Supplemental Information 19 1H- 1H COSY spectrum of compound 4

DMSO-d6, 8 MHz.

Click here for additional data file.

Supplemental Information 20 HMBC spectrum of compound 4

DMSO-d6, 8 MHz.

Click here for additional data file.

Supplemental Information 21 Structures of conformers of 4a at B3LYP/6-31G(d)level in MeOH 4a-1 34.0743 kcal/mol, 95.1827%

4a-1 34.0743 kcal/mol, 95.1827%.

Click here for additional data file.

Supplemental Information 22 Siderophores detection by using blue agar CAS assay

The CAS/HDTMA complexes were bound with ferric iron to produce a blue color as indicators. Siderophores could remove iron from the dye complex to turn it to light yellow. The light yellow ring around each colony appeared after seven days of growth indicating that P. triticisoli 1-18 had the ability of secreting siderophore and transferring iron from environment to the bacterial cells.

Click here for additional data file.

Supplemental Information 23 Untargeted metabolomics profiling of P. triticisoli BJ-18

The results of P. triticisoli BJ-18 were compared with the databases of mzCloud and Chemspider, indicating the existence of at least 101 compounds, including N_2-acetylornithine, which is the precursor of fusarinines.

Click here for additional data file.

Additional Information and Declarations

Competing Interests

Author Contributions

Data Availability

The authors declare there are no competing interests.

Yunzhi Zhang conceived and designed the experiments, performed the experiments, analyzed the data, prepared figures and/or tables, authored or reviewed drafts of the paper, and approved the final draft.

Jinwei Ren and Wenzhao Wang performed the experiments, prepared figures and/or tables, and approved the final draft.

Baosong Chen analyzed the data, prepared figures and/or tables, and approved the final draft.

Erwei Li conceived and designed the experiments, performed the experiments, authored or reviewed drafts of the paper, and approved the final draft.

Sanfeng Chen conceived and designed the experiments, authored or reviewed drafts of the paper, and approved the final draft.

The following information was supplied regarding data availability:

Raw data is available at Figshare: Zhang, Yunzhi (2020): RAW-YZ peerJ.rar. figshare. Dataset. https://doi.org/10.6084/m9.figshare.11996661.v1.

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
