# Peer review of "Siderophore and indolic acid production by Paenibacillus triticisoli BJ-18 and their plant growth-promoting and antimicrobe abilities"

_PeerJ, doi:10.7717/peerj.9403_

## Round 0.1 · original submission · Major Revisions

Unfortunately, our reviewers found important deficiencies in your manuscript: poor language throughout, lack of complete NMR spectra and excessive speculation. In my opinion, this work also requires a more complete characterization of your compounds (for example, Fe3+-complexation constants of the siderophores) and a better discussion of the possible importance of your other secondary metabolites (for example, would an analysis of possible effects of the indolic acids in plant growth be feasible?)

Finally, any further revision should take into account that a different organism ( DSM 24997) has a priority claim on the "Paenibacillus beijingensis " name. The strain used in this manuscript is DSM 25425, which was initially published as Paenibacillus beijingensis in 10.1007/s10482-013-9974-5 and changed to "Paenibacillus triticisoli" in an erratum published as 10.1007/s10482-013-0110-3

·

Basic reporting

The manuscript has many apparent inconsistencies and needs a thorough revision. English grammar should be checked throughout the text. Honestly, it is difficult to judge the quality of the manuscript itself. The language and structure are poor. The introduction section is poorly organized. The novelty and objectives of this study are not clear. The materials and methods section is not accurately written. I feel like the authors have not revised the manuscript. There are even inconsistencies with the abbreviations, acronyms and culture media composition.
The hypothesis is not presented. I would request the authors to update the literature review. In short, I can not judge the scientific quality of this article because of the above-mentioned issues.
I found this manuscript really heavy to read.

Experimental design

From my point of view, there is a clear lack of novelty in this article. The simple description of some secondary metabolites itself does no support the publication of an article. Further experiments should be conducted in order to study some specific aspects of those siderophores. The discussion section is full of expression like "it is easy to speculate...", "it´s speculate..." "it´s deduce..."
I´d suggest the authors re-write this article. Also, please focus on specific objectives/hypotheses. This article is too descriptive and the discussion is based on speculations.

Validity of the findings

No comment

Additional comments

Please avoid contractions!
Some of the acronyms are defined twice in the text (e.g. LB). Please double check.

Line 46. What part of the plants? Please be more specific
Line 48. What do you mean by "concentration"?
Lines 51-52. You said that Paenibacillus spp. can suppress the growth of plant pathogens such as bacteria, fungi, and nematodes. Do you mean that all bacteria/fungi/nematodes are pathogens? Please be more specific. You might want to add the specific pathogenic species.
Line 56. There are more recent references for that matter.
Lines 57-60. You stated that Paenibacillus was isolated from wheat. Again, be more specific. Did you mean from the rhizosphere of wheat?
Lines 70-73. We might want to define what is a siderophore and its benefits for plants.
Line 74. Why and "so on". Are not all the siderophores important? Why did you highlight those 3 of 7?
Lines 75-77. This statement is too vague. Even more, you did not define secondary metabolites which are the goal of your paper.
Lines 93-95. The objective is not clear and the hypotheses are not presented. Also, what do you mean by "multiples roles"?

The Materials and Methods section is not clear. Further details must be added. For example in Line 124 (how? concentration?), Line 141 (culture media composition), Line 147 (Incubation time? concentration?. Also, you changed from imperative to past tense to present tenses throughout this section. Please, be consistent. I found it really heavy to read.

The discussion section is full of speculations. Again, the objective is not clear and therefore your conclusions are not informative.

Reviewer 2 ·

Basic reporting

• Abstract, Intro, Methods, Results, Discussion, Conclusions are not clear and ambiguous. A thorough check of spelling and English are required.
• Figures are poorly labelled.

Experimental design

• The NMR data provided for 1 and 4 are not clear.
• Provide NMR data of 4, 5, and 6 and LC MS data for 1-6.

Validity of the findings

• Raw data provided are insufficient to determine impact/novelty of research

Additional comments

Overall, a thorough check of English is required

Reviewer 3 ·

Basic reporting

No comment

Experimental design

No comment

Validity of the findings

No comment

Additional comments

This manuscript reported six secondary metabolites from the titled bacteria, including a new siderophore which contained three fusarinine units, as well as a new paenibacillic acid A. However, the data in bioassays revealed that these compounds didn't show antimicrobial activity and cytotoxicity. Furthermore, I sugguest that the positive controls should be changed into powerful antibiotics against Staphylococcus aureus and Bacillus subtilis. The current data make no sense. Inaddition, the manuscirpt didn't provide full NMR data for compound 1, which is unacceptable for identification of a new conpund. I suggest that it should be declined.

---

## Round 0.2 · Minor Revisions

Your manuscript has improved much. Please attend to the final suggestions from our reviewers.

Reviewer 2 ·

Basic reporting

The article is clear and has sufficent references..

Experimental design

Yes the methods are sufficient.

Validity of the findings

Provide the annotated NMR spectra in the SI with the corresponding structure.

Additional comments

Comments:
Title: Siderophore and indolic acid production by Paenibacillus triticisoli BJ-18 and their plant growth-promoting and antimicrobe abilities
(pls refer to the no-markup line numbers)

Abstract
Line 25. Change “N2-fxing” to “N2-fixing”

Introduction
Line 56. Write “enzymes” in small case
Line 58. Change “referred as MoFe protein” to “referred to as MoFe protein”
Line 87. I think it is “Pirycularia oryzae” NOT “Piricularia oryzae”
Line 89. I think it is Cryphonectria parasitica” NOT “Cryphonectria parastitica”
Line 98. Insert “the” … Endophytic bacteria promote the growth…
Line 103. Change “inoculation with P. triticisoli BJ-18 significantly promote growth” to “inoculation with P. triticisoli BJ-18 significantly promotes the growth”

Materials and Methods
Line 145. I don’t understand this statement.
“The General Experimental Procedures optical rotations measurement was conducted on an Anton Paar MCP200 polarimeter (Anton Paar, Austria).”
Is “The General Experimental Procedures” a subheading?!
Line 152. Small case for “254 nm” wavelength. “UV at 254 nM”
Line 162. Use the term “seed culture” instead of “bacterial solution”
Lines 183-200. Indicate the mass error (∆) in ppm of all isolated compounds
I suggest that the numbers of the compounds are written in bold, such as compound 1, compound 2… to distinguish it from the rest of the numbers in the main text. It is very confusing.

Results
Line 270. Write ChemSpider as uppercase “S” not Chemspider
Lines 293-296. “Among the six compounds, compounds 1-3 were identified as fusarinines that were classical siderophores, while compounds 4-6 were identified as indolic acids (Fig. 2).”
Which ones are new? Among compounds 1-6? Better indicate here. I cannot follow the flow of discussion afterwards. Confusing.

Line 298. Compound 2 was a cyclic tripolymer which has three same submits (m/z 284.1366) and identified as desferritriacetylfusigen (Anke, 1977).
What do you mean submits?

The authors claimed that paenibacillic acid A and deshydroxyferritriacetyl fusigens are novel structures. Are they really novel? Or is it just a modification of known siderophores or indolic acids? If they have a completely new backbone, then the compounds are novel. Otherwise, just state as new compounds.

Line 367. Insert “the” … Effects of indolic acids compounds on the growth
Line 389. “Krber’s method” or “Spearman-Karber’s method”? Check the spelling
Line 395. Add the letter “s”…compounds 1-3 are siderophores.
Line 440. To “inhibit” the growth instead of “inhabit”
Line 448. Which supplemental files? Be specific.
Supplementary Info
Annotate each NMR spectra in the SI and show the corresponding annotated structure of each metabolite.

Figure 2. Instead of letters below the structures, use numbers 1-6 in bold similar to the the main text.
Table 3. The table caption is “Antimicrobial activity of compounds 1-6” but there are no numbers indicated in the name of the compounds.
Write the numbers 1-6 in bold after the name of the compound. What is the meaning of the blank rows and columns? Check the spelling of Karber’s method. Write as “5 µM, 10 µM and 20 µM” instead of “5 uM, 10 uM and 20 uM”

Annotated reviews are not available for download in order to protect the identity of reviewers who chose to remain anonymous.

---

## Round 0.3 · accepted · Accept

I am satisfied with your revisions.
You should indicate the specific programs used to perform the calculation of the ECD spectra. This is a minor change which I believe can be addressed with our technical team during the production stage.